# Marine-Source Quorum Quenching Enzyme YtnP to Improve Hygiene Quality in Dental Units

**DOI:** 10.3390/md19040225

**Published:** 2021-04-16

**Authors:** Xiaohui Sun, Philip Hill, Jia Liu, Jing Qian, Yuting Ma, Shufeng Zhou

**Affiliations:** 1College of Chemical Engineering, Huaqiao University, Xiamen 361021, China; liujia0505@hotmail.com (J.L.); jingqian@stu.hqu.edu.cn (J.Q.); uvtma@hotmail.com (Y.M.); szhou@hqu.edu.cn (S.Z.); 2School of Biosciences, Sutton Bonington Campus, University of Nottingham, Loughbrough LE12 5RD, UK

**Keywords:** quorum quenching, YtnP, hygiene quality, *Pseudomonas aeruginosa*, oral infection

## Abstract

Biofilm in dental unit water lines may pose a health risk to patients and dental practitioners. An AdiC-like quorum quenching enzyme, YtnP, was cloned from a deep-sea probiotic *Bacillus velezensis*, and heterologously expressed in *E. coli* to examine the application on the improvement of hygiene problems caused by biofilm infection of *Pseudomonas aeruginosa* in dental units. *Pseudomonas* bacteria were isolated from dental chair units and used to grow static biofilms in the laboratory. A water filter system was designed to test the antifouling activity of YtnP in Laboratory, to simulate the biofilm contamination on water filter in dental unit water lines. The results demonstrated that the enzyme of YtnP was able to degrade the N-acyl homoserine lactones, significantly inhibited the EPS generation, biofilm formation, and virulence factors production (pyocyanin and rhamnolipid) of *P. aeruginosa*, and was efficient on the antifouling against *P. aeruginosa*. The findings in this study indicated the possibility of YtnP as novel disinfectant reagent for hygiene treatment in dental units.

## 1. Introduction

Modern dental chair units (DCU) use water supplied via a system of tubes called dental unit water lines (DUWL) to cool their integrated instruments and for tooth irrigation during dental procedures [1]. There is an increasing awareness of the need for disinfection of DUWL as studies show counts of bacteria in water samples from DCU which are much higher than that allowed in potable water supplies [2]. Poor quality output water from DCU may be of concern in terms of health and infection control as both surgery staff and patients may inhale aerosols from the DCU instruments [3,4].

*Pseudomonas aeruginosa*, a major human opportunistic pathogen that causes oral infection and secondary diseases [5], is adapted to DCUW colonization by biofilm formation [6] and tolerance to disinfection reagents [7,8]. Studies have revealed that these resistances were regulated by the auto-inducer initiated bacterial communication called quorum sensing (QS) [9], and the auto-inducer in *P. aeruginosa* and most Gram negative pathogens were classified as N-acyl homoserine lactones (AHL) [10]. As typical characterization of QS, biofilms are bacterial communities tightly adherent to surfaces and embedded in an extracellular matrix which aids bacterial survival and makes the bacteria difficult to beremoved [11]. Bacteria growing in the biofilm mode are also difficult to kill as they exhibit increased resistance to most biocidal reagents compared to the same bacterial species growing as planktonic cultures [12].

As the supply water for DCU is normally of potable standard, it is generally accepted that the higher counts are attributable to sloughed off bacteria which are growing as biofilms in the DUWL [13]. The standard procedure of flushing with water cannot fully remove the biofilms, especially at the inlets/outlets of instruments where the physical features will affect the removal. The disinfection by using antibiotic [14,15] might cause drug resistance of pathogens [7], and the usage of chemical reagents such as glutaraldehyde [16] would be harmful to the patients and dental staff [17]. Therefore, the development of advanced antimicrobial strategies [15], such as quorum quencher, to manage oral biofilm infection [18] were highly interested by the researchers.

In previous work, an AidC-like N-acyl homoserine lactonase (EC:3.1.1.81), YtnP with novel structure and wider suitability to the well studiedAiiA from *B. thuringiensis* (PDB ID: 3DHB), had been identified as QQ enzyme from a *Bacillus velezensis* strain DH82 isolated from underlying sea water of Western Pacific Yap trench at the depth of 6000 m. In this study, the enzyme is applied to the treatment against Gram negative pathogens to verify the ability of QQ to AHLs and the application on improving hygiene quality in dental unit.

## 2. Results

### 2.1. Construction of Engineered Enzyme and Protein Expression

The expression clone of His-tagged YtnP was constructed on the plasmid of pET28a and induced for expression in *E. coli* BL21, the sequence of plasmid was confirmed by sequencing by Sangon Biotech (Shanghai) Co., Ltd.

The heterologously expressed proteins were analyzed by SDS-PAGE as shown in Figure 1, the His-tagged YtnP at 35.82 kDa was observed in lane 4, whereas the negative control of non-loaded pET28a vector in lane 2, and the crude enzyme extraction in lane 3. The concentration of harvested purified YtnP measured as 0.513 mg/mL by the Bradford assay.

### 2.2. Identification of Quorum Quenching Activity on AHL Degrading

The recombinant enzyme was firstly identified by the ability on AHL degrading against a series of auto-inducer signal in Gram negative bacteria, 3-O-C6-HSL, C6-HSL, 3-OH-C4-HSL, C4-HSL, 3-O-C12-HSL and 3-O-C10-HSL. The fluorescence intensity generated from the reporter strain was used to determine the residual AHL level after treatment in the co-culture. As shown in Figure 2, the engineered YtnP presented capacity of AHL degrading to all the tested signal molecules

### 2.3. Inhibition on Biomass Accumulation of P. aeruginosa

The QQ enzyme was applied to the bacterial culture to assess the effect of YtnP to the biomass accumulation of *P. aeruginosa.* As shown in Figure 3, the treatment of YtnP presented no significant difference on the growth curve in planktonic status (Figure 3A) but significantly depressed the gene expression of *aglD* (Figure 3B), which was related to polysaccharide synthesis and regulated the biofilm formation. The results of biofilm assessment in Figure 3C also verified the inhibition of YtnP on bacterial interruption of biofilm status, that the addition of YtnP significantly inhibited the biofilm formation of *P. aeruginosa* (*p*-value was 0.026).

### 2.4. Inhibition on Toxic Products Release in P. aeruginosa

The virulence factors were assessed to evaluate the effect of YtnP on toxin production during the growth of *P. aeruginosa*. As shown in Figure 4, the addition of YtnP remarkably reduced the release of rhamnolipid (Figure 4A) and pyocyanin (Figure 4B), which demonstrated the capacity of QQ enzyme on hygiene quality control against the Gram negative pathogen such as *P. aeruginosa*.

### 2.5. Laboratory Test of QQ Enzyme on Hygiene Quality Improvement

The effect of YtnP on the biofilm formation of *P. aeruginosa* and the permeability of filter membrane were tested on the simulated water filter system. As shown in Figure 5A, treatment of the QQ enzyme significantly inhibited the biofouling of *P. aeruginosa* on the PTEE membrane, and the experimental group treated with a free YtnP solution retained about 63.34% membrane flux after 3-day incubation, while the membrane without the QQ enzyme was rapidly blocked by the biofilm formed by *P. aeruginosa* in 2 days and remained only 6.05% on capacity of permeability after 3 days. The SEM images of the fouling layers on simulated water filter system, as shown in Figure 5B,C, were also prove the antifouling capacity of YtnP against *P. aeruginosa*. The results demonstrated that QQ enzyme treatment could be used as an effective strategy to relieve bacterial contamination and biofilm pollution on water filter in dental units.

## 3. Discussion

For some years there has been concern about the quality of output water from DCU which can be attributed to biofilm in the DUWL [19]. The biofilm bacteria may be delivered to the DUWL by poor quality input water or by contamination of reservoir bottles. Once biofilm is established in the DUWL the bacteria in it replicate and slough off into the output water, simply flushing with water is inadequate to remove the biofilm [13]. A number of disinfectant products have been developed to address the problem of DUWL biofilm with varying degrees of success [20]. The potential application of QQ enzyme had been interested as the advantage of specific targeting to interrupt the pathogens’ communication by degrading the QS signals [21].

The mechanism of QS and the relative signal molecules have been widely studied. Among the different QS signal molecules reported so far, AHLs molecules and the corresponding QQ lactonases, such as AiiA, that hydrolyze the homoserine lactone ring (HSL) are the best studied [22], and applied for biocontrol in various of areas such as agriculture [23], aquaculture [24] and water treatment [25,26]. AHLs also mediated QS of oral pathogens and caused biofilm infection in dental unit [27,28], therefore the development of QQ enzymes as novel disinfection reagents for DUWL is worth expecting as they are non-toxic and not harmful to human body and non-corrosive to the dental instruments.

*P. aeruginosa* was one of the most common pathogens isolated from the dental units, and was typical on AHLs mediated QS, including extracellular polymeric substances (EPS) producing, biofilm forming, virulence factor releasing [29], and production of an exopolysaccharide Psl for additional protection from other co-cultured antibiotic-sensitive “non-biofilm-producing” bacteria in the community [30]. In this study, *Pseudomonas* bacteria were isolated from DCU and used to assess the application of YtnP in the laboratory. The engineered YtnP was cloned from a potential probiotic strain isolated from deep-sea water, which showed no harm to the host according to previous studies. The effects of YtnP treatment during bacterial growth in rich culture media, and biofilm status under oligotrophic condition, both demonstrated the positive inhibition on bacterial EPS generation, biofilm formation and virulence factor release, by degrading AHLs to regulate the hierarchical QS system of *P. aeruginosa*. The results of laboratory tests in the simulation system of water filter also present the direct interference of YtnP to the bacterial mixture of *P. aeruginosa* on biofilm adhesion, which indicated a feasible application as disinfectant reagent in dental units, as the reduced EPS and loose biofilm structure in bacterial community would be no doubt easier for the further performance of deep cleaning by water flushing. Even though the further production of this novel disinfectant is about to carry on and trial test under international standards such as ISO 16954:2015, and relationship with other pathogens such as *Legionella spp*. from dental units [31,32,33], the findings in this study proved the possibility of YtnP as functional ingredient of hygiene disinfectant and biofilm removal reagent.

## 4. Materials and Methods

### 4.1. Bacteria, Plasmids and Reagents

100 μL aliquots of output water from five DCUs at three local dental practices were spread plated onto nutrient agar (Oxoid) and incubated at 22, 30 and 37°C. Colonies displaying different morphologies were subcultured and subjected to identification by 16s rDNA sequencing and API strips (Biomerieux). A derivative of the laboratory strain of *P. aeruginosa* PA01, and the dental isolates were used in biofilm experiments.

*B. velezensis* DH82 strain (GenBank: MK203035) was isolated from the sea water samples of the Western Pacific Yap trench at the depth of 6000 m, and was kindly offered by the Third Institute of Oceanography (Xiamen, China). The competent cell of *E. coli* DH5α and *E. coli* BL21 strain, Isopropyl-β-D-thiogalactopyranoside (IPTG) and Kanamycin were purchased from Transgen (Beijing, China). The AHL reporter operon of LuxR-P_luxI-lacO_-RFP was provided by Xiamen University (Xiamen, China). N-Acyl-L-homoserine lactone hydrolase (PDB: 3DHB) from *B. thuringiensis* (GenBank: AY943832), named AiiA_3DHB_, was used as the positive control to analyze the QQ activity of AiiA_DH82_.

The plasmid pET28a expression vector was from Novagen (Cat. 69864-3). The plasmid miniprep kit (Cat. GMK5999) and gel extraction kit (Cat. D2500-02) were purchased from Promega.The AHLs, C6-HSL (Cat. 56395), 3-O-C6-HSL (Cat. K3255), C4-HSL (Cat. 09945), 3-OH-C4-HSL (Cat. 74359), 3-O-C10-HSL (Cat. O9014), and 3-O-C12-HSL (Cat. O9139), were purchased from Sigma-Aldrich.

### 4.2. Gene Cloning

The sequence of YtnP was amplified by PCR from the genomic DNA of DH82 using the primer*ytnP*-F (5′-GGAATTCATGGAGACATTGAATATTGGGAATATTTC-3′) and *ytnP*-R (5′-CGGGATCCTTATTTTTTCTCCCGTTTGACAGATG-3′) were synthesized by Sangon Biotech (Shanghai) Co., Ltd., and was digested by the restriction enzymes *Bam*HI and *Eco*RI, and subsequently ligated the multiple cloning sites into pET28a with the T4 ligase (Takara, China). The engineered expression clone, which was driven by T7 promoter, and framed with 6× Histidine (His) at both N- and C-terminal of target enzyme, was transferred in *E. coli* BL21 for protein expression.

The operon of LuxR-P*_luxI-lacO_*-RFP was provided by Xiamen University (Xiamen, China), and was conducted on pET28a between the restriction sites of *Nde*I and *Hpa*I. The re-engineered reporter plasmid was then transferred in *E. coli* BL21 (named LuxR- RFP for short) to construct reporter strain for rapid detection of AHLs.

### 4.3. Bacterial Culture and Protein Expression

The bacteria was cultured in Luria–Bertani (LB) media (10 g/L Tryptone, 5g/L Yeast extract and 10g/L NaCl, pH7.0) with shaking at 180 rpm at 37 °C, 0.2 mM IPTG was inoculated after 2-h incubation of bacterial culture to induce the protein expression. The overnight cultured bacteria were washed and concentrated 5:1 with PBS (pH 7.0), performed ultrasonic breaking (3 s× 6 s at 300W for 60 times) on ice and centrifuged at 4 °C at 10,000 rpm for 15 min to harvest the crude extract from supernatant. The extracted crude enzyme was resuspended using a lysis buffer (300 mM NaCl and 50mM NaH_2_PO_4_ (pH 7.4)), then washed with an imidazole elution buffer (300 mM NaCl, 200 mM imidazole, and 50mM NaH_2_PO_4_ (pH 7.4)). High affinity NI-NTA chromatography was used to purify the His-tagged enzyme. The purified enzyme was filtered sterilized and analyzed by SDS-PAGE. The concentration of protein was qualified using the Bradford assay.

### 4.4. In Vitro Rapid Assessment of AHLs Level

The AHL level were determined by the relative fluorescence intensity that generated by the co-cultured reporter strain, which was calculated by dividing the above fluorescence intensity by OD600 value.

100 µL of filtered sterilized AHLs at concentration of 800mM were respectively added with 100 µL of purified enzyme for treatment. The enzyme and AHL were mixed evenly and reacted at 28 °C for 45 min, and added with 10% SDS immediately to stop the reaction. The reporter strain carrying LuxR-P*_luxI-lacO_*-RFP was incubated overnight at 37 °C with shaking at 200 rpm, and then inoculated with above enzyme-AHL mixture, incubated at 25 °C with shaking at 180 rpm for 8 h, centrifuged at 8000×*g* at 4 °C and resuspended with equal volume of PBS. The fluorescence intensity of the suspension was measured by fluorescence spectrophotometer at 620 nm (excitation wavelength at 584 nm), and the optical density (OD) at 600 nm was measured by Tecan Infinite M200 Pro. Each experiment was repeated in triplicate.

### 4.5. Assessment of Hygiene Quality on P. aeruginosa

#### 4.5.1. Growth Curve of *P. aeruginosa*

The *P. aeruginosa* overnight culture was diluted to OD600 value at 0.1, then 1:100 (*v*/*v*) inoculated in 40mL fresh LB broth for incubation at 30 °C for 12 h, with additional YtnP at final concentration of 50 μg/mL for treatment, and same volume of sterile water as negative control. The bacterial density at OD600 was measured to obtain the growth curve.

#### 4.5.2. Analysis of Gene Expression by RT-qPCR

RNA samples were respectively extracted from 1 mL supernatant of *P. aeruginosa* culture at 2, 4, 6 and 12h using Trizol RNA extraction kit (Takara), then used as template for reverse transcription by PrimeScript RT reagent Kit with gDNA Eraser (Takara) to synthesis cDNA for further qPCR program. The primer *algD*-F (5′-CACTCCAGCCGTTTCGAACT-3′) and *algD*-R (5′-CGGCTTGAACACCACCGTAT-3′); GAPDH-F (5′-GCAAGTGATGGTGGAGCAAAA-3′) and GAPDH-R (5′-CCGAGCTACTTGTGTTGGACG-3′) were respectively used to amplified the sequence of algD and GAPDH by qPCR using Bio-Rad CFX Connect.

#### 4.5.3. Assessment of Released Virulence Factors

The *P. aeruginosa* culture treated with 50 μg/mL YtnP were incubated at 30 °C for 12 h and centrifuged at 8000× *g*/min for 5 min to collect the supernatant containing the released virulence factors (pyocyanin and rhamnolipid).

0.9 mL above supernatant was mixed through with 0.54 mL chloroform, and was allowed to stand for 5 min to collect the supernatant, which was then mixed with 0.2 mL HCl. Consequently, the absorbance of the upper liquid was measured at 520 nm so as to determine the content of pyocyanin.

1 mL above supernatant was mixed with 4 mL sulphuric acid–anthrone solution (0.2 g anthrone in 100 mL 85% sulphuric acid), and was incubated in boiling water for 15 min. The content of rhamnolipid was determined by the absorbance at 620 nm.

#### 4.5.4. Microplate Biofilm Assay by Crystal Violet

*P. aeruginosa* was cultured in 200 μL of Biofilm medium (BM, filter sterilized tap water supplemented with 5 mM sodium citrate, 0.5% casamino acids (Difco) and 0.5% *v*/*v* standard brain heart infusion broth (Difco)) in 96 well microplates, with additional 50 μg/mL YtnP solution in triplicate wells for treatment, compared with sterilized water as negative control. Bacterial cultures were grown statically at 30 °C for 3 days when a luxuriant biofilm was apparent. After this time the wells were subsequently washed with phosphate buffered saline (PBS). 200 μL of 0.1% crystal violet solution was then added to each well and left for 15 min at room temperature, after which the crystal violet was removed prior to three washes with PBS. 200 μL of ethanol was then added to each well to dissolve any crystal violet bound to the well and any remaining biofilm. After 15 min at room temperature the absorbance of the wells was measured at 595 nm using Tecan Infinite M200 Pro.

### 4.6. Laboratory Test of Biofilm on Water Filter

A simulation system of water filter for biofilm contamination was assembled to quantify the biofouling of *P. aeruginosa* on the filter membrane, which consisted of a centrifuge tube and a vacuum suction filtration device connected to a catheter with a flow rate controller. The 0.22 μm PTEE filter membrane, pretreated with ultraviolet ray disinfection for 30 min, was placed and fixed steadily between the filter cup and the sand-core funnel. *P. aeruginosa* cultures of PAO1 and dental isolates in BM were separately mixed with 50 μg/mL QQ enzyme and the negative control of sterile water, and were then continuously flowed into the filter cup at about 6 drops/min under the control of a flow controller, and then flowed through the PTEE filter membrane under using gravity. For 3 d, the bacteria were intercepted so as to accumulate biofilms on the PTEE membrane.

### 4.7. SEM Imaging of Biofilm on Filter Membrane

Each PTEE filter membrane was washed with PBS three times, and then fixed with 2.5% glutaraldehyde solution at 4 °C for 2 h. The fixed samples were successively dehydrated with 25%, 50%, 75%, 95% (*v*/*v*) ethanol for 15 min each, and finally with 100% ethanol for 30 min. The dehydrated samples were immediately transferred to a vacuum oven for drying at 55 °C. The dried membranes underwent sputter coating with a gold layer and were imaged with a field emission scanning electron microscope (SEM, SU5000, Hitachi, Japan) at 8 kV.

### 4.8. Statistic Analysis

The experiments were designed in triplication; the data of which were analyzed by the software GraphPad Prism 6 and P value from T-test were used to determine the difference between each experimental group.

## 5. Conclusions

In this study, the YtnP homologous enzyme was cloned from a potential probiotic bacteria, *B. velezensis* (DH82 strain), isolated from the 6000-m deep subsurface of the Western Pacific Yap trench, and was heterologously expressed in order to investigate its QQ ability on AHL degrading, bacterial interference on biofilm contamination and virulence release, and antifouling on the laboratory simulation system of water filter. By interrupting the QS of the pathogen via degrading the AHL, the enzyme YtnP was observed to provide significant inhibition on EPS generation, biofilm formation, and virulence factor production (including pyocyanin and rhamnolipid) of *P. aeruginosa*, therefore inhibiting the bacterial fouling of *P. aeruginosa* in DUWL. The findings indicate that the QQ enzyme of YtnP could be used as an effective hygiene disinfectant reagent to prevent and control the microbial contamination of AHL-mediated Gram negative pathogens and to reduce the risk of biofilm infection by the opportunistic pathogen *P. aeruginosa* in dental units.

## Figures and Tables

**Figure 1 marinedrugs-19-00225-f001:**
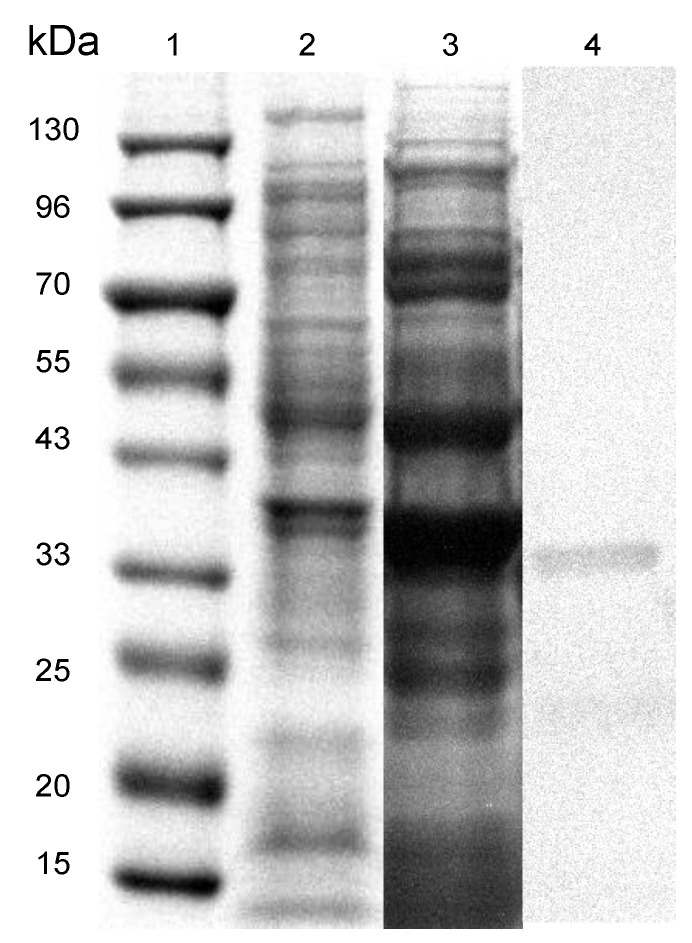
SDS-PAGE analysis of engineered YtnP. The predicted quorum quorum quenching enzymes YtnP was expressed overnight at 37 °C in *E. coli* under induction with 0.4 mM IPTG, the expressed proteins were harvested and analyzed by SDS-PAGE. Lane 1, marker; Lane 2, negative control of pET28a vector; Lane 3, crude extraction from bacteria; Lane 4, purified YtnP enzyme.

**Figure 2 marinedrugs-19-00225-f002:**
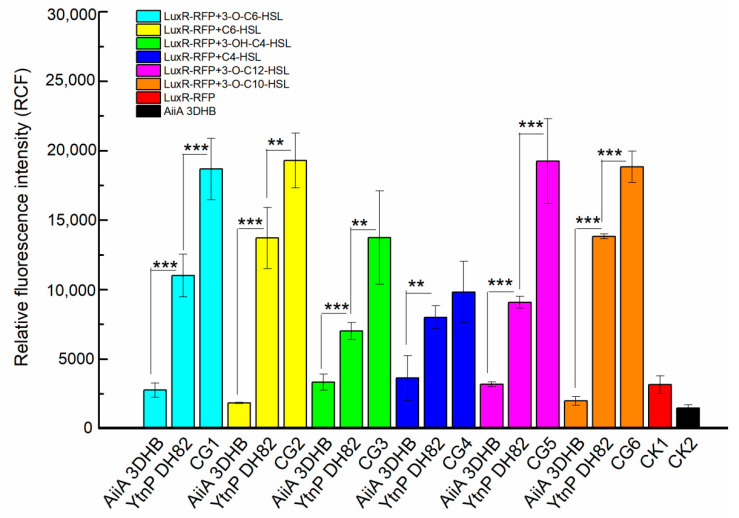
Activities of quorum quenching enzymes on AHLs degrading.The degradation activities of YtnP against a series of AHLs were respectively present. Six experimental groups of reporters added with the different AHLs (3-O-C6-HSL in cyan, C6-HSL in yellow, 3-OH-C4-HSL in green, C4-HSL in blue, 3-O-C12-HSL in purple and 3-O-C10-HSL in orange) that respectively treated with crude YtnP extraction, comparing with that treated with AiiA3DHB as positive control, and the untreated AHLs as control groups (CGs). The control checks of reporter without AHL (CK1 in red) and the AiiA3DHB only (CK2 in black) were also present. Statistical analysis results are presented by significant difference that indicated by *** where *p* < 0.01 and ** where *p* < 0.05

**Figure 3 marinedrugs-19-00225-f003:**
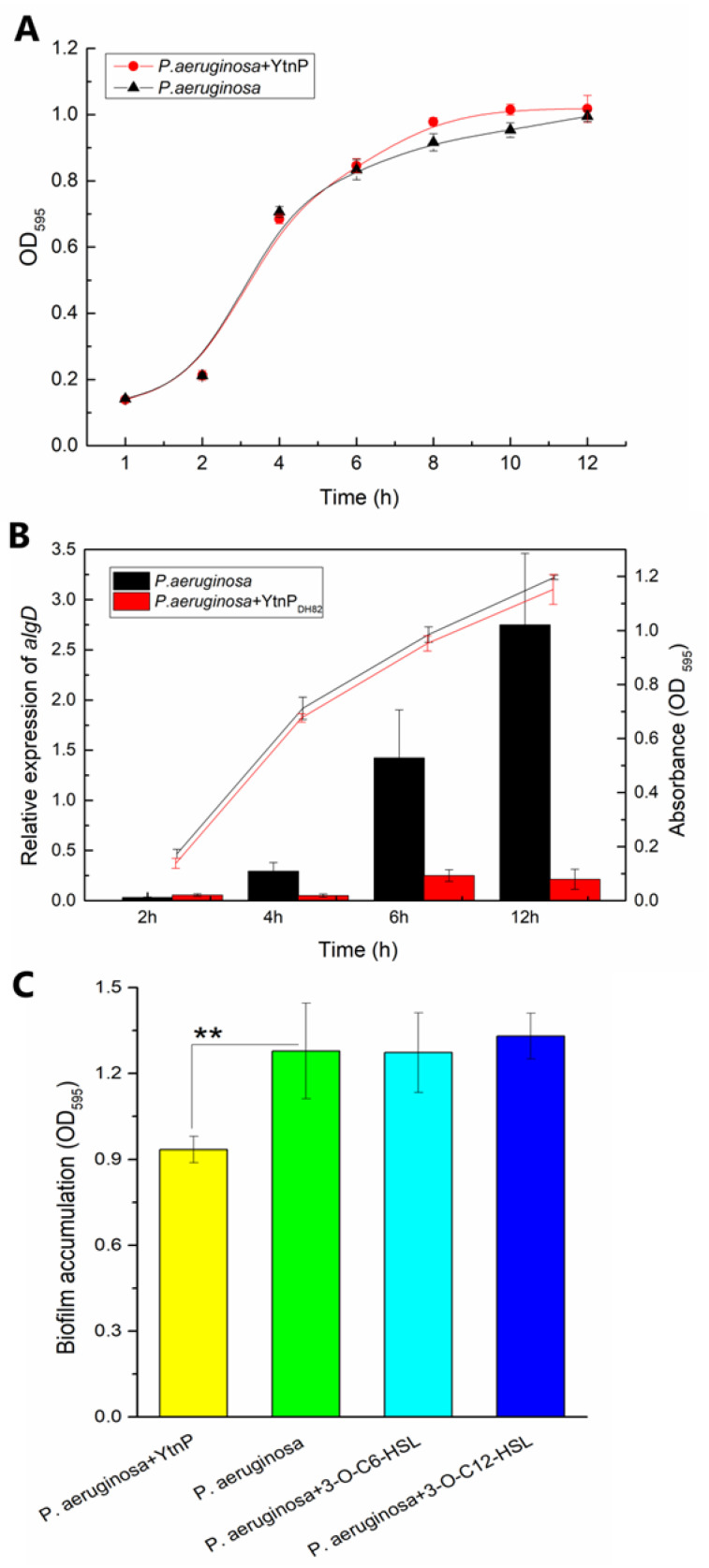
Effect of YtnP on biomass accumulation of *P. aeruginosa.* YtnP was inoculated to the bacterial culture of PAO1 for treatment, the bacterial density for growth in LB and the biofilm accumulation in BM were determined by absorbance at 595 nm. (**A**) shows the growth curve of *P. aeruginosa*; (**B**) shows the relative gene expression of *aglD* during bacterial growth using GAPDH coding gene as reference; (**C**) presents the biofilm accumulation under the treatment of YtnP (in yellow) comparing with that of non-treatment (in green) and addition of 3-O-C6-HSL (in cyan) and 3-O-C12-HSL (in blue). Error bars are used to determine the standard deviation. Statistical analysis results with a significant difference are marked using as significant differences at 0.01< *p* < 0.05 indicated by **.

**Figure 4 marinedrugs-19-00225-f004:**
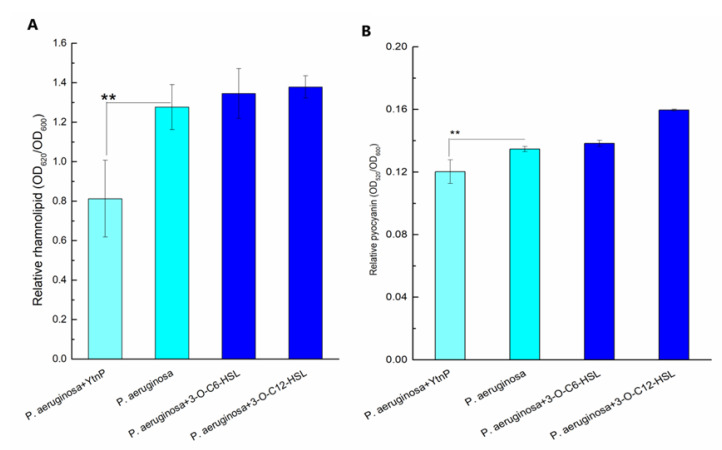
Toxic products release during bacterial growth.50 μg/mL YtnP was inoculated to the bacterial culture of PAO1 for treatment, the bacterial density, amount of released pyocyanin and rhamnolipid were determined by absorbance at 600, 520, and 620 nm, respectively, measured using a microplate reader. (**A**) shows relative rhamnolipid and (**B**) shows relative pyocyanin in bacterial culture (bacterial culture with enzyme treatment in light cyan, non-treated bacterial culture in cyan, and bacterial culture with the addition of AHLs in navy). Error bars are present to determine the standard deviation. Statistic analysis results with significant difference indicated at 0.01< *p* < 0.05 as **).

**Figure 5 marinedrugs-19-00225-f005:**
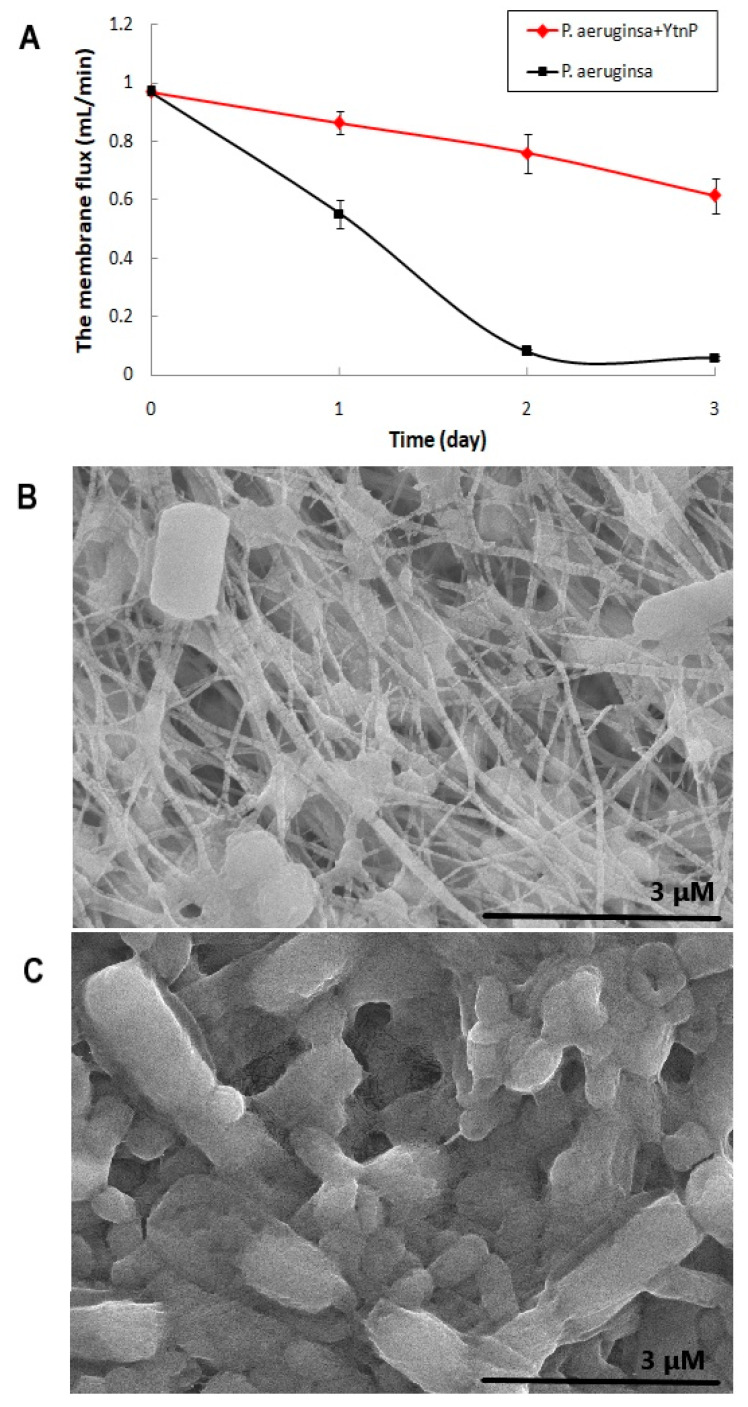
Effectof YtnP on antifouling on PTEE filter membrane.*P. aeruginosa* PAO1 and dental isolates were inoculated and grown in BM with additional 50 μg/mL YtnP in the simulated water filter system, bacterial culture were continuously pumped through a 0.22 μm PTEE filter membrane for 3 d. (**A**) shows the permeability of filter membrane with or without treatment of YtnP, where the permeability was determined by the flux of sterile water that flowed through the PTEE filter membrane per minute(mL/min). The error bars present the standard deviation. (**B**) was SEM image of biomass accumulation on PTEE membrane treated with YtnP at 15,000× magnification. (**C**) was SEM image of contaminated non-treated PTEE membrane at 15,000× magnification.

## Data Availability

The data presented in this study are available on request from the corresponding author.

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
