# Peer review of "Marine-Source Quorum Quenching Enzyme YtnP to Improve Hygiene Quality in Dental Units"

_marinedrugs, 2021, doi:10.3390/md19040225_

Round 1

Reviewer 1 Report

The authors aimed to clone, the YtnP homologous enzyme from a potential probiotic bacteria, B. velezensis (DH82 strain), isolated from the 6000-m deep subsurface of the Western Pacific Yap trench, in order to investigate its QQ ability on AHL degrading, bacterial interference on biofilm contamination and virulence release, and antifouling on the laboratory simulation system of water filter.

The structure of the manuscript appears adequate and correctly divided in sub-paragraphs. The methodology is well described with enough experimental data and results to support the work. Please check English and typos thorough the text.

Conclusion Section: This paragraph required a general revision to eliminate redundant sentences and to add some "take-home message".

Author Response

Thanks for the comment and suggestion. The authors had read through the manuscript and correct the wrong typing and English mistakes.  

Reviewer 2 Report

The paper entitled "Marine-source quorum quenching enzyme YtnP to improve hygiene quality in dental units" is an original paper which investigated the inhibitory effects on quorum sensing among bacteria colonizing the dental unit, with the aim to reduce the risks of contamination and biofilm formation in the DUWL . In detail, they found a significative inhibitory effect on biofilm formation of P.aeruginosa and its toxic products .

Abstract is concise and adequate.

Introduction is well written and exhaustive.

Please, control line 46 and 51 for square brackets missing at the reference n.13 and 15, respectively. I suggest authors to re-check the manuscript for similar typing errors.  

Figures and schemes well support and enrich the presentation of results. The SEM images in Fig. 5 are very suggestive and well presented.

M&M section  is sufficiently detailed and the methods used are adequate for the purposes.

Discussion is clear and well written. In order to depict the “state of the art” on DUWL issues, I suggest authors to read and refer the following papers:  

Carinci F, Scapoli L, Contaldo M, Santoro R, Palmieri A, Pezzetti F, Lauritano D, Candotto V, Mucchi D, Baggi L, Tagliabue A, Tettamanti L. Colonization of Legionella spp. In dental unit waterlines. J Biol Regul Homeost Agents. 2018 Jan-Feb;32(2 Suppl. 1):139-142. PMID: 29460533.

Tuvo B, Totaro M, Cristina ML, Spagnolo AM, Di Cave D, Profeti S, Baggiani A, Privitera G, Casini B. Prevention and Control of Legionella and Pseudomonas spp. Colonization in Dental Units. Pathogens. 2020 Apr 21;9(4):305. doi: 10.3390/pathogens9040305. PMID: 32326140; PMCID: PMC7238104.

Ditommaso S, Giacomuzzi M, Ricciardi E, Memoli G, Zotti CM. Colonization by Pseudomonas aeruginosa of dental unit waterlines and its relationship with other bacteria: suggestions for microbiological monitoring. J Water Health. 2019 Aug;17(4):532-539. doi: 10.2166/wh.2019.240. PMID: 31313992.

The conclusions are adequate and the future prospect of the study could encourage further studies on the development and application of these "biological systems" to reduce the use of harmful antibiotics and disinfectants, as suggested by the authors.

In conclusion, the paper is well written and I suggest to revise minor typing errors along the text and to spend few words on other literature reporting the DUWL colonization and their different approach to disinfection.

Author Response

Thanks for the comment and suggestion. The authors had read through the manuscript and correct the wrong typing and English mistakes. The suggested paper had been cited in the article to enhance the description in discussion section.

Reviewer 3 Report

This paper presents some very  interesting data describing experiments that utilize a quorum quenching enzyme from a marine organism to mediate and improve water quality in dental units.  I found the experiments described extremely interesting  and potentially of some commercial value.  I recommend its publication in your journal  

Author Response

Thanks for the comment and suggestion about accepting the manuscript for publication.